# Early Treatment of Unilateral Condylar Hyperplasia in Adolescents: Preliminary Results

**DOI:** 10.3390/jcm12103408

**Published:** 2023-05-11

**Authors:** Sergio Olate, Victor Ravelo, Juan Pablo Alister, Henrique Duque Netto, Ziyad S. Haidar, Roberto Sacco

**Affiliations:** 1Division of Oral, Facial and Maxillofacial Surgery, Dental School, Universidad de La Frontera, Temuco 4780000, Chile; 2Center of Morphological and Surgical Studies (CEMyQ), Universidad de La Frontera, Temuco 4780000, Chile; 3Grupo de Investigación de Pregrado en Odontología (GIPO), Universidad Autónoma de Chile, Temuco 4810101, Chile; 4Department of Oral and Maxillofacial Surgery, Federal University of Juiz de Fora, Juiz de Fora 36000-000, Brazil; 5Centro de Investigación e Innovación Biomédica (CiiB), Universidad de los Andes, Santiago 7550000, Chile; 6BioMAT’X R&D&I (HAiDAR I+D+i) LAB, Facultad de Odontología, Universidad de los Andes, Santiago 7550000, Chile; 7Division of Dentistry, Oral Surgery Department, School of Medical Sciences, The University of Manchester, Manchester M13 9PL, UK; 8Oral Surgery Department, King’s College Hospital NHS Trust, London SE5 9RS, UK

**Keywords:** facial asymmetry, unilateral condylar hyperplasia, orthognathic surgery, TMJ

## Abstract

Facial asymmetry associated with unilateral condylar hyperplasia (UCH) is a rare disease. The aim of this study was to evaluate the clinical conditions of progressive facial asymmetry in young subjects treated with high condylectomy. A retrospective study was performed including nine subjects diagnosed with UCH type 1B and progressive facial asymmetry around 12 years old with an upper canine progressing towards dental occlusion. After an analysis and a decision of treatment, orthodontics began one to two weeks prior to the condylectomy (with a mean vertical reduction of 4.83 ± 0.44 mm). Facial and dental asymmetry, dental occlusion, TMJ status and an open/closing mouth were analyzed before surgery and in the final stage of treatment, almost 3 years after surgery. Statistical analyses were performed using the Shapiro–Wilk test and a Student’s t-test considering a *p* value of <0.05. Comparing T1 (before surgery) and T2 (once orthodontic treatment was finalized), the operated condyle showed a similar height to that observed in stage 1 with a 0.12 mm difference in height (*p* = 0.8), whereas the non-operated condyle showed greater height increase with an average of 3.88 mm of vertical growth (*p* = 0.0001). This indicated that the non-operated condyle remained steady and that the operative condyle did not register significant growth. In terms of facial asymmetry in the preoperative stage, a chin deviation of 7.55 mm (±2.57 mm) was observed; in the final stage, there was a significant reduction in the chin deviation with an average of 1.55 mm (±1.26 mm) (*p* = 0.0001). Given the small number of patients in the sample, we can conclude that high condylectomy (approx. 5 mm), if performed early, especially in the mixed-dentition stage before full canine eruption, is beneficial for the early resolution of asymmetry and thus the avoidance of future orthognathic surgery. However, further follow-up until the end of facial growth is required.

## 1. Introduction

Facial asymmetry associated with unilateral condylar hyperplasia (UCH) is rare. It has been confirmed that surgical treatment using high condylectomy, proportional condylectomy or low condylectomy is useful in treating the disease [1,2,3].

The diagnosis of UCH requires analyses such as clinical study, clinical photography, follow-up with computed tomography (CT) and single-photon emission computerized tomography (SPECT) [4]. The low specificity and sensitivity of SPECT and the negative impact of frequent irradiation in childhood and adolescence—if used for the follow-up of condylar growth—could be a limitation in the regular use of SPECT [5]; complementary clinical and image analyses are necessary.

The incidence of mandibular condylar hyperplasia has been associated with adolescents and young women [6,7], showing a hormonal role in the disease; thus, early adolescence could be a key point in UCH. Between 10 and 15 years of age, the mandibular condyle is strong in the growth process [8], which is associated with mandibular development and aesthetic balance [9,10]. In this sense, the psychological and social impacts on adolescents in terms of facial morphology are significant [11].

The early treatment of UCH using proportional condylectomy does not commonly require a secondary orthognathic surgery [3], and early treatment could improve facial growth balance [12].

On other hand, facial asymmetry as a result of non-controlled UCH could be complex to treat. Lopez et al. [13] included six different types of facial asymmetry, showing differences in the morphology of the mandible or maxillae with differences in size and proportions. This complexity was considered by Kwon et al. [14], who showed differences in soft tissue after 1 year of surgery in subjects with previous facial asymmetry and demonstrated the challenge of treating facial asymmetry. This conclusion was recently supported by van Riet et al. [15], who detailed the protocol to treat facial asymmetry including an analysis in facial contouring. Severe facial asymmetry is complex to treat, so early treatment can help achieve a good standard in facial balance.

The aim of this study was to analyze the result of early condylectomy in subjects with UCH treated at 12 years old, at the same time their upper canines were erupting.

## 2. Materials and Methods

This study was conducted with the patients’ consent and the informed consent of their parents or guardians. This study was conducted in accordance with the guidelines of the Declaration of Helsinki and approved by the Ethics Committee for Research with the protocol UFJF-IRB 2.148.583.

Nine consecutive subjects with progressive facial asymmetry were diagnosed with type 1B UCH (defined by Wolford et al. in 2014 [1] as unilaterally abnormal condylar growth with a relatively normal architecture of the condyle and the enlargement of the condylar head; the condylar neck tends to thicken while the mandibular ramus tends to grow in height vertically). Female and male patients were included who were between 11.4 and 13.1 years old with upper canines in the process of eruption. Subjects who presented previous facial trauma, complex facial pathology or previous facial surgery were excluded.

The diagnosis of unilateral condylar hyperplasia was obtained by clinical and image criteria including:(1)Patient and family records which indicated mandibular deviation and progressive asymmetry in the last year.(2)Clinical dental study: unilateral cross bite and deviation of the interincisal midline (Figure 1).(3)Facial analysis: deviation of the chin by more than 5 mm from the facial midline, falling within mandibular class III, with or without maxillary cant (Figure 2).(4)Cone beam computed tomography (CBCT): evidence of mandibular condyles of greater volume and size; vertical measurements of the mandibular ramus showing differences (asymmetry) between the right and left sides (Figure 3)(5)SPECT: presence of a positive SPECT study with differences of 10% between the captured image of the two condyles. Differences lower than 10% were included according to the full analysis of the case.

Between one and three weeks before the surgery, orthodontic appliances were installed to take care of the dental occlusion after the condylectomy; the location of the bracket and slots was defined by the orthodontist.

The surgery was performed under general anesthesia. With a previously described technique [16], an endaural approach was used to access the joint capsule, exposing the mandibular condyle; the osteotomy was performed using a piezoelectric system. The measurement to obtain the level of the osteotomy in the condylar head was based on a proportional condylectomy [17]. In most cases, the proportional condylectomy was close to 5 mm, so a high condylectomy could be applied as well (Figure 4) (Table 1). In cases with extensive dissection over the condylar neck, the dissection of the lateral pterygoid muscle was necessary. The initial dissection was performed in the anterior area of the condylar head and neck using a distraction of the condyle with mandibular movement by the assistant surgeon. Then, a posterior approach was used with an open mouth position to enable the greater visibility of the entire structure, after which an osteotomy was performed.

Condylectomy produces a mandibular setback on the surgical side and a new upper position of the operated condyle due to the effect of the masseter and temporal muscles; this creates premature dental contact on the operated side mainly at the level of the first or second molar (Figure 5). This clinical condition has been considered previously; after surgery, an anterior open bite is expressed, usually with a normal overjet.

Two to three days after surgery, the use of elastics began, with the goals of centering the midline and of achieving a class I canine using moderate force (Figure 5). Between three and five days after surgery, physiotherapy was started with facial muscle treatment, lymphatic drainage and finally, joint mobility.

The follow-up monitored the progress and movement of the mandibular component and guided dental occlusion. Orthodontic treatments were extended from 24 to 35 months after the surgery. The follow-up using CBCT was conducted at the end of the orthodontic treatment, taking the effects of irradiation on the patients into consideration.

The variables included were the presence of a dental midline, chin symmetry, the size of the mandibular condyles, dental condition and molar class, TMJ activity in opening and closing movements, the presence of pain and postoperative joint noise.

Linear measurements of the mandibular condyle were taken using the Ez3D Viewer Plus software (Vatech, Yongin, Republic of Korea). During the measurement of the condylar unit (Table 2), two further images were obtained for each type, immediately before and immediately after (1 mm cuts), and a final number was obtained from the average of the three measurements in the anteroposterior view, mediolateral view and condylar neck length (Table 2); the measurement was performed by one operator twice in two week intervals. The correlation coefficient between the measurements was 0.91 (*p*-value of 0.001).

The data analysis was performed with Graph Prism v. 9.1.0. The clinical parameters are presented as means (X) and standard deviations (SD). The Shapiro–Wilk test was used for the analysis of the normal distribution. For the evaluation and comparison of the continuous variables before and after the condylectomy, a Student’s paired t-test was used. For the correlation only, between the preoperative variables, a Pearson’s test was used with consideration of a value of *p* of < 0.05 as a significant difference.

## 3. Results

Nine patients between 11 and 13 years of age were diagnosed with type 1B UCH. In every case, progressive facial asymmetry was confirmed by patient and family description, clinical study and a 3D image. The subjects were treated with high condylectomy and orthodontics with no postoperative complications (Figure 7).

In all the patients, a class III angle was confirmed at the molar level on the UCH side. A class I, II or III angle was noted on the side unaffected by UCH; moreover, a trend of class III facial deformity growth was observed. After surgery, an open bite was observed on the non-operated side and premature contact at a molar level was observed on the operated side. Physiotherapy was started in the first week after surgery (Table 3 and Table 4).

In the SPECT registry, a difference of less than 10% was observed in two subjects. The clinical analysis during diagnosis showed that facial asymmetry was present with a chin deviation of 7.55 mm (±2.57 mm) with an interdental midline deviation of an average of 5.44 mm (±1.91 mm). The differences in the height of the affected condyle and the condyle unaffected by the UCH was 4.83 mm (±0.44 mm); the difference in the width of the condyles was 2.12 mm (±0.78 mm). There were significant differences between the two condyles (*p* = 0.007) (Table 5).

The observations made in T2, once the orthodontic treatment was complete, between 24 and 35 months after the surgery, noted that the operated condyle showed a similar height to that observed in T1 with a 0.12 mm difference in height (*p* = 0.8), whereas the non-operated condyle showed greater height increase with an average of 3.88 mm of vertical growth (*p* = 0.0001) (Figure 8). The operated condyle was in recovery, and the recovery obtained on the top of the condyle was almost the same vertical condition as in T1. In T2, the non-operated condyle was bigger than the operated condyle with no clinical change in terms of dental occlusion or facial asymmetry.

A similar condition occurred with regard to the width of the condyles, with no differences in the width of the operated condyle between T1 and T2 (*p* = 0.06) and with significant differences in the width of the non-operated mandibular condyle (*p* = 0.001) (Table 6).

In the last CBCT evaluation, the repair of the operated mandibular condyle was observed. In terms of facial asymmetry, in the preoperative stage, a chin deviation of 7.55 mm (±2.57 mm) was observed. In the final, stage there was a significant reduction in chin deviation with an average of 1.55 mm (±1.26 mm) (*p* = 0.0001) (Figure 9). Class I dental occlusion was noted in every case, and a dental midline was confirmed as well. No open bite or changes in dental occlusion were observed.

## 4. Discussion

Between 10 and 16 years old, almost 70% of the mineralization of the cortical bone of the mandible is complete [18]. An uncontrolled disease in the mandibular condyle under 16 years old can cause alterations in the growth and development of the mandible [19]. Genetic conditions associated with UCH as well as phenotypes play a role in this scenario [20].

Wolford et al. [1] identified UCH in adolescents, also of type 1B, defined as the unilaterally abnormal growth of the mandibular condyle with a relatively normal architecture of the condyle and an enlargement of the condylar head. The condylar neck usually increases in thickness and the vertical height of the mandibular ramus also increases. The condition is different in adolescents compared to adults, because age has an important role in the disease; it has been reported in previous studies that the comparison of subjects of different ages could affect their conclusion [21].

Nolte et al. [22] included 148 subjects, with UCH with 80% of them undergoing surgery, of an average age of 20 years and ranging from 9 to 54 years of age. Mouallem et al. [23] operated on 73 patients, with UCH, of an average age of 22 years and a range from 10 to 58 years. Fariña et al. [3] included subjects of an average age of 19 years in their study, whereas Slotwed and Muller [24] included 22 subjects to develop their histological classification, where the patients were of ages of over 14 years. This difference could explain some conclusions related to the diagnosis and techniques of treating UCH patients.

Our patients were treated with the protocol of early condylectomy and orthodontic procedures. After orthodontic treatment, a stable condition was observed in terms of dental occlusion and maxilla-mandibular position; however, as the patient is in active growth, changes in development may be visible. Therefore, these patients must maintain regular contact with the surgeon and orthodontist until the growth process is complete.

A condylectomy can repair and act on condylar movement after surgery [25,26]. Abbound et al. [27] indicated that condylectomy can delay the advance of the disease, but not of the three-dimensional defect in the face; however, these authors treated subjects between 17 and 25 years, with no chance of changing their facial morphology. Aerden et al. [12] reported that condylectomy performed on subjects with skeletal immaturity (14 years ± 1.73) improved the dental midline from 2.44 mm to 1 mm of deviation and changed the bony midline from 3.22 mm to 1.67 mm, and that only 22% required secondary orthognathic surgery. Our results showed a significant change in the facial midline with an average deviation of 1.5 mm in the final evaluation, where orthodontics had a great role; only two 17-year-old female patients required genioplasty corrections to improve the proportion and facial aesthetics. Orthognathic surgery has not been performed on any patient to date; however, after final growth, the real requirement of orthognathic surgery will be evaluated.

Di Blasio et al. [28] used panoramic X-rays and clinical analyses in their follow-up. They concluded that condylectomy can correct condylar anatomy, and that the growth of the patient can improve the final symmetry. The morphology of the operated condyle in our patients takes the new condylar shape. In T2, the non-operated condyle was bigger than the operated condyle, changing the anatomical condition with no change in facial symmetry or dental occlusion.

However, a fully proportional condylectomy in adolescents may imply the risk of reverse asymmetry at the end of growth (i.e., depending on the growth potential left). Therefore, the primary aim of a condylectomy in adolescents should be growth arrest, taking into consideration the growth potential to be expected from the contralateral side when defining vertical resections.

In this sense, mandibular condyles under endochondral growth may modify the anatomical structure. In studies with animal models, it has been observed that the installation of growth factors contributes to the increase in volume in the growing condyle [29]. On other hand, it has been observed that environmental elements such as mechanical load and facial deformities can accelerate changes in the morphology of the mandibular condyle [30,31]. Our observations show that early condylectomy can bring about a new shape and morphology of the condylar head. The adaptations made to the function and the growth of the facial skeleton also contribute to the process [25,26,32].

The “wait and see” option adopted in some UCH cases could be related to facial growth with an asymmetric pattern in the entire facial morphology. The UCH causes significant facial changes in the hard and soft tissues [33], which are difficult to fully correct. In our clinical practice, we use the “wait and see” strategy in some cases, and this may be an option for physician and patients; however, in this sample of patients with progressive facial asymmetry, early changes in dental occlusion and facial asymmetry were observed after surgery, showing stable results in the beginning. Thus, this is an option to obtain better conditions in adolescents in terms of appearance and psychological concerns.

In this clinical series, the clinical decision was to perform early interventions to minimize the impact of the disease on young people. Post-condylectomy function has been studied by some authors [34,35], and they have demonstrated high effectiveness with positive results and few complications. In our sample, no complications in the regular function of the TMJ was observed when the orthodontic treatment was finished; however, the absence of measurements in mandibular movements such as laterotrusive or protrusive movements make drawing conclusions in this area difficult.

Two patients in our sample were treated with SPECT differences under 10%. In some studies [2,3], the performance of condylectomy on subjects with negative SPECT was explained by progressive facial asymmetry. SPECT is used to assess diagnoses in UCH, but it is not without controversy [5]. Despite the negative SPECT results in two cases, we opted for surgery in line with the diagnosis described in M&M (family records, dental and clinical conditions, and CBCT study) due to the progression of the asymmetry. In these cases, the involvement of facial appearance, psychological implications and the complexity of achieving regular dental occlusion using exclusive orthodontic appliances early on helped us decide on surgery as the best option in accordance with patient, parents and clinical criteria.

The results presented in this study are interesting, but caution is needed because of the limitations inherent in a case series, such as the small sample size, the lack of full measurements in terms of condylar movement, the use of exclusive linear measurements in the condyle (with no superimposition), the limitation in the 3D follow-up of the condyle immediately after surgery, and the lack of follow-up until the patients finished growing, which close to when they were 18 or 19 years old. We expect the patients to continue growing, and follow-ups will be required in the final growth stage.

## 5. Conclusions

High condylectomy (approx. 5 mm), if performed early, especially in the mixed-dentition stage before full canine eruption, is beneficial for the early resolution of asymmetry and thus the avoidance of future orthognathic surgery.

## Figures and Tables

**Figure 1 jcm-12-03408-f001:**
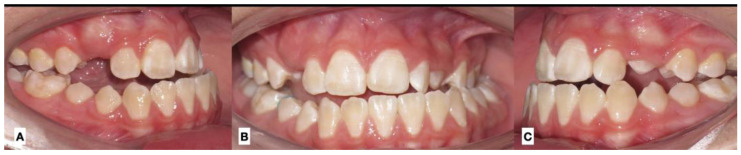
Dental occlusion at the initial evaluation: (**A**) right view, (**B**) frontal view, and (**C**) left view. Unilateral crossbite and progressive change in dental midline. Upper canine in progress.

**Figure 2 jcm-12-03408-f002:**
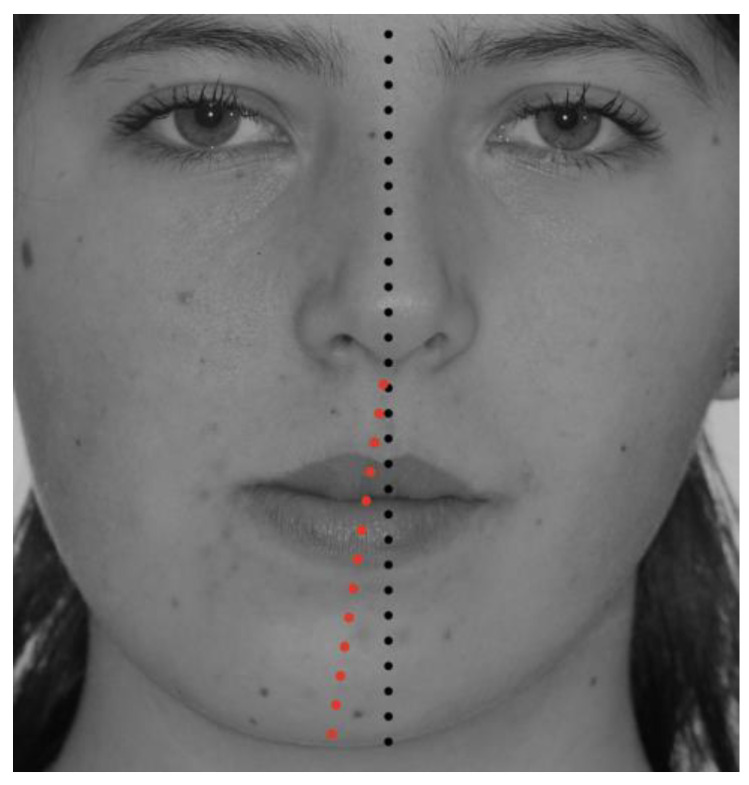
Facial midline with a deviation of 6 mm to the left, showing progressive facial asymmetry related to condylar hyperplasia of the right mandibular condyle.

**Figure 3 jcm-12-03408-f003:**
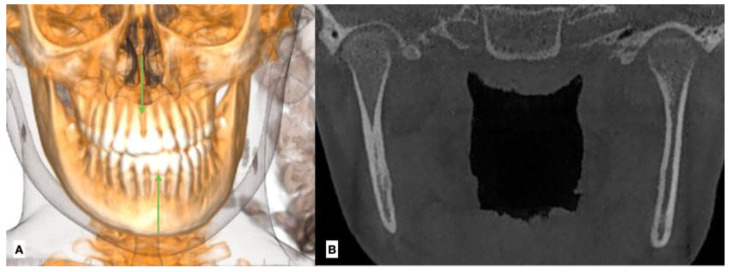
(**A**) Facial condition showing asymmetry and change between upper and lower dental midline (green zeta); (**B**) CBCT shows greater size and volume in the right condyle, creating a movement of the midline to the left side.

**Figure 4 jcm-12-03408-f004:**
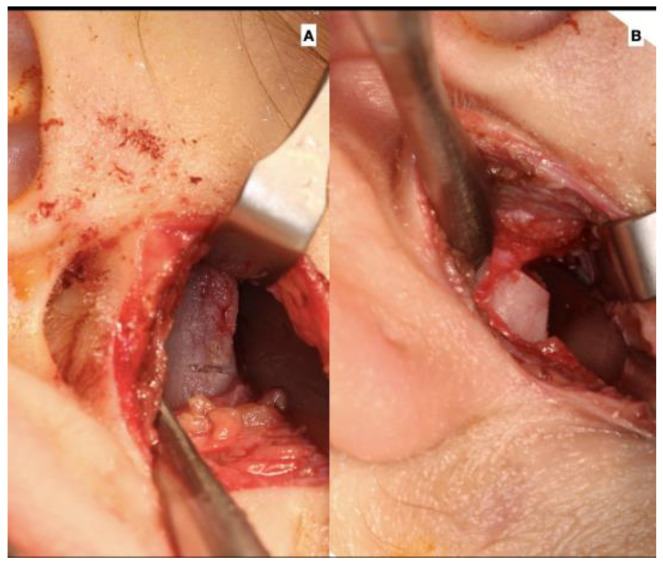
Surgical approach for TMJ (**A**) with the approach and observation of the right condyle and (**B**) condylar osteotomy performed with piezo surgery.

**Figure 5 jcm-12-03408-f005:**
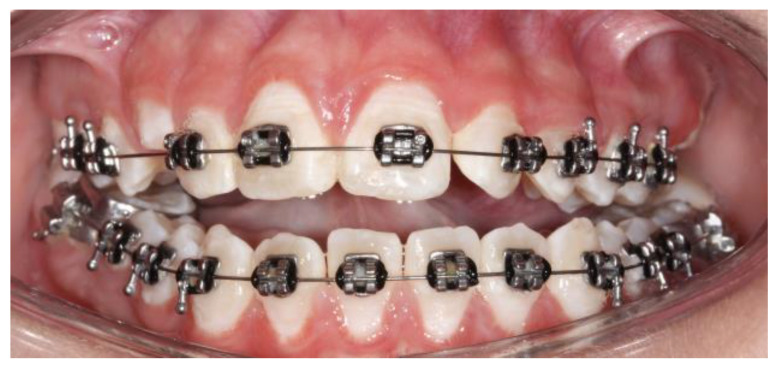
Anterior open bite obtained after condylectomy with primary dental contact on the surgical side (right mandibular condyle in this case).

**Figure 6 jcm-12-03408-f006:**
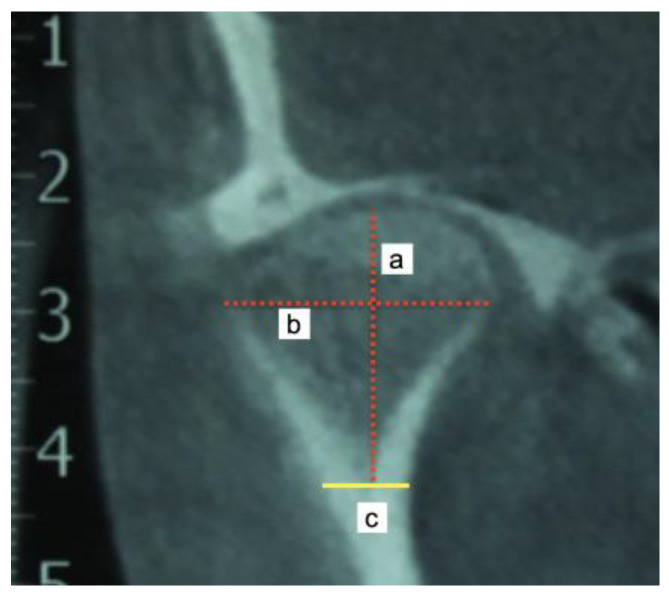
Measurements obtained in the condylar head: (a) mandibular condyle height, (b) mandibular condyle width, and (c) lower limit of the condylar neck.

**Figure 7 jcm-12-03408-f007:**
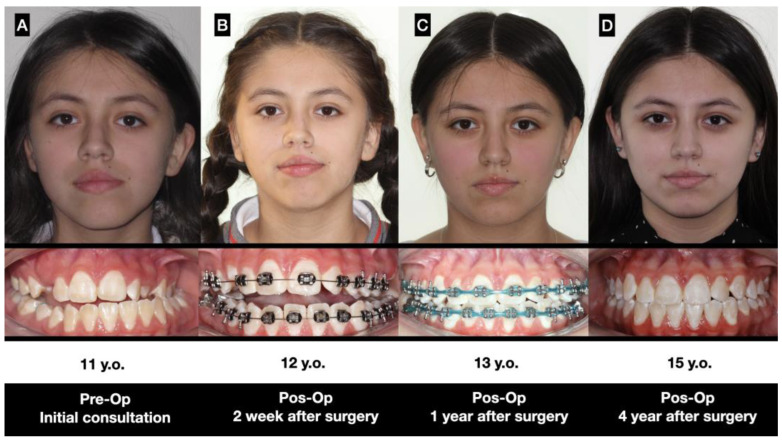
Patient treated under this protocol. (**A**) Initial analysis with facial and dental asymmetry, (**B**) 2 weeks after surgery with the classic condition of an anterior open bite and the use of 1/8″ and 3/16″ inch elastics, (**C**) 1 year after surgery wirth better facial balance and stabilized dental occlusion, and (**D**) 4 years after surgery without braces with occlusion stability and a stable dental and facial midline.

**Figure 8 jcm-12-03408-f008:**
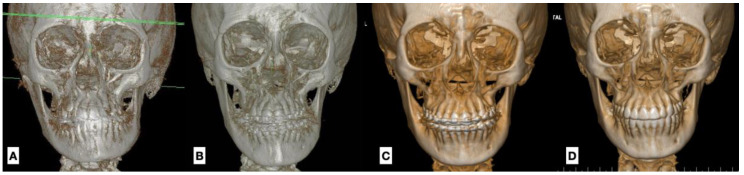
Evolution of the facial skeleton in unilateral condylar hyperplasia (**A**) during the initial diagnosis, (**B**) after condylectomy with the better condition of the mandible midline, (**C**) 1 year after surgery showing a better balance in the midline and in the ramus and mandibular body size and morphology, and (**D**) at the end of treatment showinggood symmetry in dental position and the facial skeleton; however, the chin position is still deviated to the left and genioplasty could be necessary.

**Figure 9 jcm-12-03408-f009:**
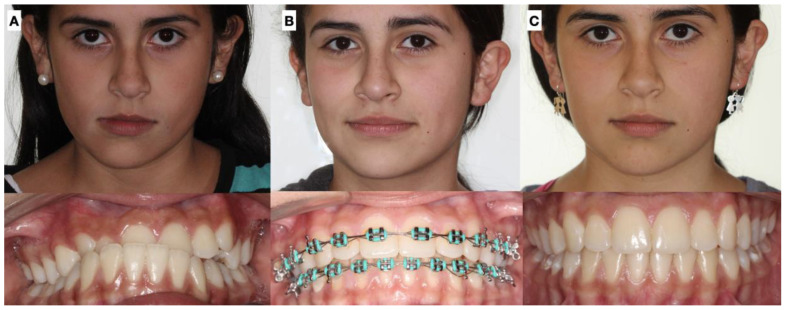
Patient treated under this protocol. (**A**) Progressive facial asymmetry and asymmetric class III trend, (**B**) 1 year after condylectomy showing stability in facial and dental midline, (**C**) 4 years after surgery with stability in facial condition and class I dental occlusion, and right deviation of the chin with the patient refusing reposition by short genioplasty.

**Table 1 jcm-12-03408-t001:** Description of the type of condylar osteotomy in UCH.

High condylectomy	Osteotomy in the condylar head removing 5 mm from the top of the condyle	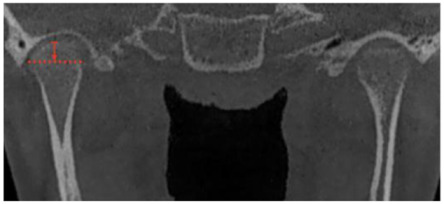
Low condylectomy	Osteotomy in the condylar head removing the entire condylar head up to the neck	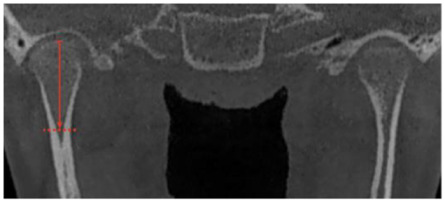
Proportional condylectomy	Osteotomy in the condylar head removing the necessary millimeters from the top of the condyle to obtain an equal height between the right and left ramus-condyle units. No quantity or anatomical area is mandatory to be removed as in the high or low condylectomy techniques.	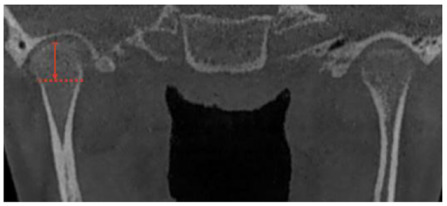

**Table 2 jcm-12-03408-t002:** Description of the measurement performed by CBCT for analysis of the condylar head.

Measurement	Description
Mandibular condyle height	Coronal view. A longitudinal line from the uppermost cortical point of the condylar head to the lower limit of the condylar head (division with the condylar neck) (Figure 6)
Mandibular condyle width	Coronal view. A longitudinal line at the widest point of the condyle on the axial axis of the condyle, starting and ending at the closest point of the most medial and lateral cortical bone. (Figure 6)
Condylar head length	Coronal view. A longitudinal line at the flat point of the condyle, usually in the lower landmark of the cancellous bone of the condylar head
Dental midline	Lack in continuity between the upper dental midline and the lower dental midline. Measurement obtained in millimeters between the upper and lower difference (Figure 1)
Facial midline	Difference between the facial midline (obtained from the glabella and pronasale) and the midline of the chin (Figure 2)

**Table 3 jcm-12-03408-t003:** Characterization of the nine patients included in this research before condylectomy (T1).

ID	UCH Side	SPECT R	SPECT L	SPECT Dif.	Mandibular Condyle Height			Mandibular Condyle Width			Dental Midline	Facial Midline
					UCH+	UCH−	Dif.	UCH+	UCH−	Dif.		
1	L	44	56	12	18.5	13.5	5	16.1	14	2.1	4	7
2	R	62	38	24	20.1	14.2	5.9	15.9	13.8	2.1	5	6
3	L	43	57	14	18.7	14.1	4.6	17.1	15.5	1.6	5	7
4	R	53	47	6	19.6	15	4.6	16.2	14.2	2	5	8
5	L	42	58	16	19.8	15	4.8	16.3	14.2	2.1	6	8
6	L	45	55	10	18.1	13.1	5	17.2	13.1	4.1	7	9
7	L	41	59	18	18.5	14.0	4.5	16.8	14.9	1.9	6	9
8	L	39	61	22	17.9	13.2	4.7	14.2	12.3	1.9	5	7
9	L	46	54	8	18.6	14.2	4.4	14.9	13.6	1.3	6	7

Note: T1. UCH+: mandibular condyle with condylar hyperplasia; UCH−: mandibular condyle with no condylar hyperplasia; R: right side; L: left side.

**Table 4 jcm-12-03408-t004:** Characterization of the nine patients included in this research after orthodontic treatment (T2), between 25 and 35 months after surgery.

ID	UCH Side	Mandibular Condyle Height			Mandibular Condyle Width			Dental Midline	Facial Midline
		UCH+	UCH−	Dif.	UCH+	UCH−	Dif.		
1	L	19.5	18.5	1	16.4	15.2	1.2	1	0
2	R	20.5	18.2	2.3	16.1	16.8	-0.7	0	2
3	L	19.2	18.5	0.7	17.4	15.9	1.5	2	2
4	R	20.1	19.0	1.1	16.4	16.2	0.2	0	0
5	L	16.2	15.0	1.2	15.9	16.6	-0.7	1	3
6	L	19.3	17.1	2.2	17.7	17.1	0.6	2	2
7	L	19.2	18.9	0.3	17.1	15.9	1.2	0	2
8	L	18	18.2	-0.2	14.7	14.3	0.4	1	3
9	L	18.9	17.8	1.1	15.2	14	1.2	0	0

Note: T1. UCH+: mandibular condyle with condylar hyperplasia; UCH−: mandibular condyle with no condylar hyperplasia; R: right side; L: left side.

**Table 5 jcm-12-03408-t005:** Comparison of the preoperative height and width of mandibular condyle of the nine subjects included in the study.

	Mandibular Condyle Height		Mandibular Condyle Width		*p* Value
	X	SD	X	SD	
Affected condyle (UCH+)	18.86	7.66	16.07	6.56	0.007 *
Unaffected condyle	14.03	4.48	13.95	4.50	
Average	4.83		2.12		

Note—T1: preoperative stage; X: average of measurements; SD: standard deviation. * indicates a statistically significant difference.

**Table 6 jcm-12-03408-t006:** Comparison of the height and width of the mandibular condyle before the surgery and after orthodontic treatment.

	T1		T2		*p* Value
	X	SD	X	SD	
Mandibular condyle height					
Affected condyle (UCH+)	18.86	7.66	18.98	7.62	0.8
Unaffected condyle	14.03	4.48	17.91	5.78	0.0001 *
Mandibular condyle width					
Affected condyle (UCH+)	16.07	6.56	16.32	6.66	0.06
Unaffected condyle	13.95	4.50	15.77	5.09	0.001 *
Facial midline					
Chin asymmetry	7.55	2.57	1.55	1.26	0.0001

Note—T1: preoperative stage; T2: end of orthodontic treatment. X: average of measurements; SD: standard deviation. (*) indicates a statistically significant difference.

## Data Availability

The data are available upon request from the corresponding author.

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
