# Peer review of "Early Treatment of Unilateral Condylar Hyperplasia in Adolescents: Preliminary Results"

_jcm, 2023, doi:10.3390/jcm12103408_

Round 1
Reviewer 1 Report (Previous Reviewer 2)
Since condylar hyperplasia is a rare disease, I agree more data and information is required for this field. I happily see this article. I have two minor requests.
1. From the tables 3 to 6, several measurements are presented. Mandibular condyle height / width, dental midline and facial midline. I couldn’t find the definitions of these measurements. Please describe the definitions and how authors measured them (I would like to recommend presenting figures)
2. Measurement’s reliability should be validated with method error estimation. For example, same examiner re-measure the values and statistically (such as intraclass correlation) evaluate them with the first measured values.
Author Response
Dear reviewer
thank you very much for your comments
- Since condylar hyperplasia is a rare disease, I agree more data and information is required for this field. I happily see this article. I have two minor requests.
R: Thank you
- From the tables 3 to 6, several measurements are presented. Mandibular condyle height / width, dental midline and facial midline. I couldn’t find the definitions of these measurements. Please describe the definitions and how authors measured them (I would like to recommend presenting figures)
R: Thank you for the comment. Were included a change in table with description of measurements and a new figure showing the measurements.
- Measurement’s reliability should be validated with method error estimation. For example, same examiner re-measure the values and statistically (such as intraclass correlation) evaluate them with the first measured values.
R: Error method was realized in all the measurements. The data was no included at the beginning, however, in this review was included a statement with a new paragraph in the M&M.
Reviewer 2 Report (New Reviewer)
Thank you for the opportunity to review the paper "Early Treatment of Unilateral Condylar Hyperplasia to Resolve Class III Progressive Facial Asymmetry".
This is a very interesting paper focusing on facial and dental morphological analysis following early surgical treatment of condylar hyperplasia
Please see below for specific comments and editing suggestions.
The title of the pdf document does not match the online title:
"Early treatment of unilateral condylar hyperplasia in adolescents. Preliminary results" & "Early treatment of unilateral condylar hyperplasia to resolve progressive class III facial asymmetry"
What is the correct title?
In the material and method, it is mentioned the realization of 2 CBCT
"Cone beam computed tomography (CBCT): evidence of mandibular condyle of greater volume and size; vertical measurements of the mandibular ramus show differences (asymmetry) between the right and left sides (Fig 3)" & "The follow-up using CBCT was done at the end of the orthodontic treatment, taking the irradiation for the patients into consideration. Figure 7 shows 4 CBCTs, please explain.
It is a pity that the authors have evaluated only the morphological aspect. TMJ has an important functional role. The authors wrote in the discussion section: "In our sample, no alteration in TMJ was observed when the orthodontic treatment was finished; however, the absence of measurements in mandibular movements such as laterotrusive or protrusive movements make concluding in this area difficult", without numerical results, it is not possible to write: "no alteration in TMJ was observed". Please modify or provide data in the results section. For this reason, "morphologic" should possibly be added to the title
Author Response
|
The title of the pdf document does not match the online title: "Early treatment of unilateral condylar hyperplasia in adolescents. Preliminary results" & "Early treatment of unilateral condylar hyperplasia to resolve progressive class III facial asymmetry" What is the correct title? R: the title is Early Treatment of Unilateral Condylar Hyperplasia in Adolescents. Preliminary Results; this is different from the submission because another reviewers asked to change in the process. For that reason, there is a change in title.
In the material and method, it is mentioned the realization of 2 CBCT. "Cone beam computed tomography (CBCT): evidence of mandibular condyle of greater volume and size; vertical measurements of the mandibular ramus show differences (asymmetry) between the right and left sides (Fig 3)" & "The follow-up using CBCT was done at the end of the orthodontic treatment, taking the irradiation for the patients into consideration. Figure 7 shows 4 CBCTs, please explain. R: thank you!. Indeed, 2 cbct were performed in the process to assess the measurements; however, in the figure 7 we are showing the evolution related to one patients; for the measurements, we used only two CBCT.
It is a pity that the authors have evaluated only the morphological aspect. TMJ has an important functional role. The authors wrote in the discussion section: "In our sample, no alteration in TMJ was observed when the orthodontic treatment was finished; however, the absence of measurements in mandibular movements such as laterotrusive or protrusive movements make concluding in this area difficult", without numerical results, it is not possible to write: "no alteration in TMJ was observed". Please modify or provide data in the results section. For this reason, "morphologic" should possibly be added to the title R: thank you!. You are right; was modified the paragraph. Was changed to “no complications in the regular function” |
Reviewer 3 Report (New Reviewer)
Firstly, I would like to congratulate the authors for a very interesting work.
I think these considerations should be taken into account.
1.at materials and methods
....Nine consecutive subjects with progressive facial asymmetry were diagnosed with type 1B UCH (defined by Wolford et al. As a unilaterally condylar abnormal growth with relatively normal architecture of the condyle and enlargement of the condylar head; the condylar neck tends to thicken while the mandibular ramus tends to grow in height ver-tically). ... you should include the article reference of that definition.
2.In my opinion, I think that the conclusions ( the last 2 paragraphs) at the end of the article should be in a single paragraph and be more concise.
Author Response
- Firstly, I would like to congratulate the authors for a very interesting work.
R: Thank you
2.at materials and methods
....Nine consecutive subjects with progressive facial asymmetry were diagnosed with type 1B UCH (defined by Wolford et al. As a unilaterally condylar abnormal growth with relatively normal architecture of the condyle and enlargement of the condylar head; the condylar neck tends to thicken while the mandibular ramus tends to grow in height vertically). ... you should include the article reference of that definition.
R: The reference was included in the M&M
2.In my opinion, I think that the conclusions ( the last 2 paragraphs) at the end of the article should be in a single paragraph and be more concise.
R: Thank you, you are right. The last 2 paragraph is a discussion. Was included a last chapter of conclusion with a single paragraph.
Round 2
Reviewer 1 Report (Previous Reviewer 2)
Thanks to the authors for their revision of the article.
This manuscript is a resubmission of an earlier submission. The following is a list of the peer review reports and author responses from that submission.
Round 1
Reviewer 1 Report
Topic of this research is interesting and worth to study. However, this paper needs lot of work.
Introduction is incomplete. Writing in the chapter could be more fluent. Chapter deals with UCH but it should also include more information about facial asymmetry with relevant citations. Objective of the study should be more informative, and it should clearly state the reason for the study.
Materials and Methods is also incomplete. Sample size isn’t mentioned. There should be some description for measurable variables. With CBCT models, far more informative asymmetry measurements (3D?) could have been implemented. Paired t-test should be used when same subject is measured twice.
Author Response
Rev: Introduction is incomplete. Writing in the chapter could be more fluent. Chapter deals with UCH but it should also include more information about facial asymmetry with relevant citations. Objective of the study should be more informative, and it should clearly state the reason for the study.
Reply: was included a new paragraph related to facial asymmetry with 3 new references:
- López DF, Botero JR, Muñoz JM, Cárdenas-Perilla R, Moreno M. Are there mandibular morphological differences in the various facial asymmetry etiologies? A tomographic three-dimensional reconstruction study. J Oral Maxillofac Surg 2019; 77: 2324-38
- Kwon SM, Baik HS, Jung HD, Jang W, Choi YJ. Diagnosis and surgical outcomes of facial asymmetry according to the occlusal cant and menton deviation. J Oral Maxillofac Surg 2019; 77: 1261-75
- van Riet TCT, Klop C, Becking AG, Nolte JW. Management of asymmetry. Oral Maxillofacial Surg Clin N Am 2023; 35: 11-21
Rev: Materials and Methods is also incomplete. Sample size isn’t mentioned.
Reply: the sample size was in the results chapter and now was moved to the M & M
Rev: There should be some description for measurable variables. With CBCT models, far more informative asymmetry measurements (3D?) could have been implemented.
Reply: the measurements used in this research were the requirements for this method. More measurements can be used for new analysis, however the authors believe that the reported measurements in this case are correct to describe the aim of this research; a table included this topic was included in the main text to define the measurements and landmarks.
Rev: Paired t-test should be used when same subject is measured twice.
Reply: Thank you; this test was used in this research
Reviewer 2 Report
I believe this study is very educational and informative to the readers. Only one thing I want to recommend is adding a schematic drawing of "proportional condylecomy, high/low condylectomy for better understanding of the readers.
Author Response
dear reviewer, thank you. Was included a table with definitions and a CBCT to show the differences.
Round 2
Reviewer 1 Report
This manuscript needs lots of work. It is great that there are lot of informative figures but many time it is hard to follow the text itself. It needs repairing and just adding things is not enough.
Introduction
Nothing has been done to make introduction more fluent. It doesn’t introduce enough. There could be something general info about UCH and facial asymmetry.
There still isn’t any clear reason why this study exists.
Material and methods
Sample should be described better. Why are ages mentioned two times? How many subjects were excluded?
Description for measurable variables is way too unclear. Three measurements from table 2 are not used later. There’s a mention that a final number is average of these three. What is a final number and where is it used?
Measurements shown is results should be described properly (image would also be good) in Material and Methods.
In the line 135 of page 5, there is list variables: Why? How are these used in the paper?
One of main interest of this paper is facial asymmetry. Still, only one parameter is used, and it measures only chin asymmetry. Facial asymmetry is much more. If you study facial asymmetry, you should also measure what happens to asymmetry on the upper part of face.
Results
Information about patients (in first 2 paragraphs) should be in materials not results.
Presentation of results is too disorganized. Results of table 4 are in text before results of table 3. They are mixed in same paragraph so it’s hard to follow.
Sample size is very low, so I prefer to see also confidence intervals in the tables.
Minor correction:
Page 5 line 145: analysis of normal distribution -> test of normality
Page 5 line 150: No need to mention sample size again
Page6 line 166: What is number between brackets (+-2.57 mm)? Is it measurement error or confidence interval. It could be mentioned when using first time in the text.
Page6 line 168: 4.84 mm, but in Table 3: 4.83
Page 6 Table 3: P-value should be in the same row as the comparable averages
Page6 line 190: p=0.06 but in Table 4: 0.6
Author Response
Dear Reviewer
Thank you so much for your comments.
Rev: Introduction - Nothing has been done to make introduction more fluent. It doesn’t introduce enough. There could be something general info about UCH and facial asymmetry.
Reply: we do not understand what mean more fluent. We believe that introduction is enough to go through the main topic and include info about the problem; we added a new paragraph as you ask in the first round about facial asymmetry.
Rev: There still isn’t any clear reason why this study exists.
Reply: the aim of this research is included in the last paragraph of introduction.
Rev: Material and methods: Sample should be described better. Why are ages mentioned two times? How many subjects were excluded?
Reply: thank you; was deleted the line with the age. This is a consecutive clinical series; subjects who showed the inclusion criteria were admitted to this protocol.
Rev: Description for measurable variables is way too unclear.
Reply: the article in M&M show a paragraph “The variables included were the presence dental midline, chin symmetry, size and volume of mandibular condyles, dental condition and molar class, tmj function in opening and closing movement, presence of pain and postoperative joint noise.” The variables included for the analysis are present in the text and the results are chowed in the next chapter.
Rev: Three measurements from table 2 are not used later.
Reply: the measurements were used for condyle measurement and statistical analysis. Table is a description of measurements with landmark
Rev: There’s a mention that a final number is average of these three. What is a final number and where is it used?
Reply: a better description was included in the text.
Rev: Measurements shown is results should be described properly in Material and Methods.
Reply: table 2 show a description of measurements.
Rev: One of main interest of this paper is facial asymmetry. Still, only one parameter is used, and it measures only chin asymmetry. Facial asymmetry is much more. If you study facial asymmetry, you should also measure what happens to asymmetry on the upper part of face.
Reply. Thank you. We think that the main interest of this paper in facial asymmetry, however in these cases related to HCU (rare cases) and in the early adolescence (still more rare cases) and the authors believe that this series is a good contribution to the field because the lack of information about this pathology in young people and the promising results about the early treatment with proportional condylectomy. This a clinical series with a large follow up showing our results in consecutives patients treated by our team. we show the reparation of the condyle and a really good recovery in facial symmetry with clinical protocol and proper methodology.
We are agree with you in term of definition of facial asymmetry. In the authors opinion the measurements show a results of the surgical and dental protocol and the problem and objective of this research is cover using the proposal methodology.
Results
Rev: Information about patients (in first 2 paragraphs) should be in materials not results.
Reply: the info was moved to M&M
Rev: Presentation of results is too disorganized. Results of table 4 are in text before results of table 3. They are mixed in same paragraph so it’s hard to follow.
Reply: the paragraph was moved
Rev: Sample size is very low, so I prefer to see also confidence intervals in the tables.
Reply: sample is low because the presence of this pathology is rare. This a case series with a special young group and a large follow up.
Rev: Page 5 line 145: analysis of normal distribution -> test of normality
Reply: thank you
Rev: Page 5 line 150: No need to mention sample size again
Reply: ok, thank you
Page6 line 166: What is number between brackets (+-2.57 mm)? Is it measurement error or confidence interval. It could be mentioned when using first time in the text.
Reply: (±2.57 mm) is the standard deviation; this is mentioned in the M&M and in the tables
Page6 line 168: 4.84 mm, but in Table 3: 4.83
Reply: this is 4.83, thank you
Page 6 Table 3: P-value should be in the same row as the comparable averages
Reply: this comparison was between affected and unaffected condyle in the initial diagnosis; so, the position in the tables is to show the real differences between them at the initial moment.
Page 6 line 190: p=0.06 but in Table 4: 0.6
Reply: this is 0.06 and was modified in the table, thank you